# More than just a bad day? Traumatic life events and self-control in old age

**Youngjoo Choung[1], Tae-Young Pak[iD][2]***

**1** Department of Consumer Science, Inha University, Incheon, Republic of Korea, **2** Department of Consumer Science and Convergence Program for Social Innovation, Sungkyunkwan University, Seoul, Republic of Korea

\* typak@skku.edu

## Abstract

The behavioral economics literature suggests that exposure to traumatic events shifts preference features including risk aversion and time preference. In this study, we examined the association between traumatic life events and self-control in old age. Data were obtained from the Health and Retirement Study, which offers retrospective data on trauma exposure and early life characteristics. The results showed that experiences of serious physical attacks or assaults is associated with a 3.1% reduction in self-control, adjusted for demographic and childhood socioeconomic characteristics. The attacks or assaults were experienced approximately 30 years prior to the survey, indicating that traumatic life events exert a lasting influence on self-control. Further analyses found no difference in the association between the experience of serious physical attacks or assaults and self-control according to the timing of occurrence. Our findings are consistent with the evidence that experiences of natural disasters or armed conflicts increase impatience among survivors.

**Data Availability Statement:** Data underlying the study are third-party data publicly available from the following URL: https://hrs.isr.umich.edu/about. The authors confirm they did not have any special access to this data that others would not have.

## Introduction

Self-control is the ability to regulate "thoughts, feelings, and actions when enduringly valued goals conflict with momentarily more gratifying goals" [1]. People with strong self-control are more likely to achieve long-term goals and attain a higher level of well-being in various domains [2]. Several studies have shown that children's self-control ability predicts academic achievements and cognitive coping skills in adolescence [3,4]. Economists have modeled self-control as a parameter of "willpower" in the intertemporal choice model [5] and demonstrated its association with economic behavior and financial well-being in adulthood [6–8]. Self-control is especially important for older adults in light of the evidence that the self-control problem is linked to asset decumulation and self-assessed financial distress in old age [9].

There is growing scholarly interest in the determinants of self-control. Emerging evidence suggests that traumatic life events alter the structure and neurochemistry of the brain, thereby leading to poor self-control [10]. The human brain is known to encode external stimuli as threatening or non-threatening, prompting the person to adjust their behavior to the perceived threat level [11]. When experiencing a potentially threatening event, the brain enters a state of

**Funding:** This work was supported by the Ministry of Education of the Republic of Korea and the National Research Foundation of Korea (NRF-2021S1A5A8069342). The funders had no role in study design, data collection and analysis, decision to publish, or preparation of the manuscript.

**Competing interests:** The authors have declared that no competing interests exist.

"mental combat mode" and redirects energy and resources away from advanced reasoning and toward a primitive functioning necessary for survival [12]. Sustained exposure to threats and the related hormonal changes lead to the underactivation of the brain's modulatory functioning and undermine one's ability to resist temptation [13].

Prior research on the formation of self-control has focused on the relatively short-term effects of traumatic events experienced in early childhood [14]. Hence, it is not yet fully understood whether adolescence trauma exposure has a long-term effect and the shifts in self-control persist into later life. In the current study, we examined the association between traumatic life events and self-control in old age. Using retrospective data drawn from the Health and Retirement Study, we estimated the models that link trauma exposure in earlier life to self-control two or three decades later. The results showed that the experience of serious physical attacks or assaults is associated with a 3.1% decline in self-control approximately 30 years after the incidents. The estimated trauma effect is three times greater than the gender gradient in self-control and is roughly comparable to the effect of unemployment. This result indicates that traumatic life events exert a significant influence on self-control. This study contributes to three strands of research. First, we contribute to the research on human capital accumulation, which reports a negative association between adverse childhood conditions (e.g., illness, in-utero alcohol exposure, poor nutrition) and health and economic outcomes in later life [15,16]. Although self-control is generally not considered a form of human capital, it has been considered a valuable psychological resource that enables people to follow through on their commitment to human capital investments and achieve desirable life outcomes [17,18].

Second, our study adds to the growing literature on preference shifts after trauma exposure. Evidence that traumatic life events alter preference features, such as risk aversion and time preference, has been found in survivors of major natural disasters [19–23], terrorist attacks [24], and civil conflicts and wars [25–29], exploiting the events as a source of the causal effect of trauma. A similar finding was reported in a study of retrospective data on negative life events and risk preferences [30]. To our knowledge, no study has examined the potential association between negative life events and self-control.

Finally, this study contributes to human development research that explores the formation and evolution of self-control over life. The conventional theory states that self-control is established early in life and remains relatively stable afterward [31]. However, current knowledge is insufficient to explain the lifetime variation in self-control and associated well-being outcomes among traumatized individuals. We expand the literature by taking a longer lifetime perspective of self-control development in relation to traumatic life events.

## Previous research

### Traumatic life events as a determinant of self-control

The Diagnostic and Statistical Manual of Mental Disorders defines trauma as exposure to "actual or threatened death, serious injury, or sexual violence" [32]. Common experiences of trauma include "direct trauma exposure, witnessing trauma, or learning about trauma that happened to a close friend or relative" [33]. A memory of childhood trauma is associated with an increased risk of post-traumatic stress disorder and a higher incidence of depression, anti-social behaviors, and substance abuse [34,35].

A growing body of psychiatry research has examined the link between trauma exposure and self-control problems. Studies have found a positive association between childhood trauma and pathological impulsivity in patients with gambling disorder [36], substance abuse [37], and suicidal behavior [38]. Even in healthy populations, the memory of abuse or neglect in early childhood is associated with reduced executive functions [39], deviations in cognitive

control [40], and poor self-regulation over smoking and drinking [41] in adulthood. Psychological responses to adverse experiences and the associated emotional effects have been suggested as potential paths leading to executive dysfunction [42].

Traumatic experiences can have an adverse impact on the brain's modulatory functions. When the body senses environmental threats, the stress response system becomes activated and shifts the body's metabolic resources away from the thinking part of the brain and toward the more primitive functioning related to survival [33]. While the body fights off potential threats, advanced brain functions that are not necessary for survival become less active. As a result, the frontal lobe, the brain's decision-making center, starts functioning slowly, and its connectivity with the other parts of the brain is weakened. This process helps individuals mobilize more effectively in the event of an emergency, but it dampens their ability to make complex decisions, such as controlling impulses [43].

Several studies have revealed that traumatic experiences lead to structural changes in the brain and reduce the mental capacity for self-control [33,44,45]. Neuroimaging studies have shown that childhood trauma is associated with reduced volume in brain areas that manage memory/emotion processing and behavioral regulation [46]. Similarly, pediatric research has demonstrated that the volume of certain brain areas related to executive functions is relatively small in abused and deprived youths [47]. The symptoms of post-traumatic stress disorder can last for more than two decades after trauma exposure, leading to brain shrinkage in some key areas [48]. Experiments with animals have also shown that external stress undermines one's ability to perform complex tasks and behaviors and triggers structural changes in brain circuits [49]. Whether such changes in brain structure persist in old age remains unclear.

## Traumatic life events and economic preferences

A host of economics research has shown that negative life experiences alter risk and time preferences. For instance, Cameron and Shah [20] conducted risk game experiments with Indonesians living in disaster-prone areas and found that those who had experienced floods or earthquakes preferred less risky options. Similarly, Jakiela and Ozier [28] examined the civil conflict after the election in Kenya and showed that those who were directly exposed to the conflict exhibited high risk aversion. This upward shift in risk aversion has also been observed among survivors of tsunamis [21], drug-related violence [26], and wars [25,27,29]. In particular, Cassar et al. [21] found that the 2004 tsunami in Thailand led to an increase in impatience, prosocial behavior, and risk aversion.

Evidence also demonstrates that the preference shift may be long-lasting but not permanent. Notably, Bellucci et al. [25] and Kim and Lee [29] showed that exposure to war is linked to a shift in risk preference several decades later. Bellucci et al. [25] used childhood exposure to war as a natural experiment and found that those who lived in a region with intense battles during the war exhibited higher risk aversion in old age. Kim and Lee [29] demonstrated that the association between war intensity and risk aversion is more pronounced for those who were aged four to eight years during the war, suggesting sensitive developmental periods for preference formation. Bucciol and Zarri [30] obtained a similar finding through analyses of retrospective data, reporting that the experience of a physical attack or losing a child is correlated with reduced risky investments from the family financial portfolio.

The direction of the preference shift remains unclear, with evidence suggesting that preferences could be altered in both directions after an event. Studies like Page et al. [50] and Kahsay and Osberghaus [51] have found a downward shift in risk aversion among survivors of floods and storms. However, Abatayo and Lynham [52], Eckel et al. [22], and Hanaoka et al. [23] reported that the risk-seeking response after a natural disaster is significant only for a

particular gender. Voors et al. [53] found increased impulsivity and altruistic behavior in the aftermath of a civil war, whereas Callen [19] highlighted an increase in the monthly discount factor among tsunami survivors in Sri Lanka.

## Methods

### Data description

Data were drawn from the Health and Retirement Study (HRS), a population-based longitudinal survey of Americans over the age of 50 years living in the United States. The HRS has been conducted annually since 1992 by the Institute for Social Research of the University of Michigan. The survey is a rich source of data on the changing life circumstances that Americans undergo toward the end of their work lives and in the years after retirement. The HRS covers four broad areas: income and wealth, labor supply, health, and healthcare access. The key measures for the current study, self-control and traumatic life events, were added to the HRS in 2006 as part of the Psychosocial Leave-Behind (LB) module. The LB module was administered to a randomly selected pool comprising 50% of the sample alternatingly so that each participant responded to the module every four years. Traumatic life events were fielded in 2006, 2008, 2010, and 2012 waves, and self-control was fielded in the 2010 and 2012 waves. The response rates for the LB module were 73.1% and 72.7% for the 2010 and 2012 waves, respectively.

The study sample consisted of respondents aged 50–80 years in the 2010 and 2012 HRS. The sample was restricted to individuals with valid measurements of self-control, traumatic life events, and other covariates. The final sample included 8,767 person-level observations (4,714 from the 2010 wave and 4,053 from the 2012 wave). Our estimates were weighted by the sample weights that reflect the complex sampling procedures of the HRS and LB module.

**Measure of self-control.**   The Multidimensional Personality Questionnaire (MPQ) is a factor-analytically developed self-report instrument [54]. The MPQ is designed to measure 11 primary personality traits, one of which is self-control. The self-control subscale consists of six items measuring one's mental capacity to override impulses and take a long-term perspective. The respondents were asked to rate their ability in the domains of money management (Q1: "I keep close track of where my money goes"), attention to tasks (Q2: "I often stop one thing before completing it and start another"), action with deliberation (Q3: "I often act without thinking"), tendency to forecast (Q4: "Before I get into a new situation, I like to find out what to expect from it"), alertness (Q5: "I am often not as cautious as I should be"), and planning ahead (Q6: "I often prefer to play things by ear rather than to plan ahead") using a six-point Likert scale, ranging from "strongly disagree" (1) to "strongly agree" (6). Negatively phrased items (Q2, Q3, Q5, and Q6) were reverse-coded such that a higher summary score represents greater self-control. A summary score was obtained from the mean of the six responses, resulting in a continuous score of 1–6. High scores indicate that a participant is particularly careful, reflective, or planful and thus has great self-regulation ability. The internal consistency of the scale was acceptable ($\alpha = 0.78$).

**Measures of traumatic life events.**   Traumatic life events were assessed using the lifetime trauma inventory [55]. The traumatic events considered in this study include (a) a serious physical attack or assault ("physical attack" hereafter); (b) a major fire, flood, earthquake, or other natural disaster ("natural disaster" hereafter); and (c) a life-threatening illness or accident ("life-threatening illness" hereafter). For each event, respondents were asked to indicate whether and when they had experienced the suggested traumatic event. Using these responses, we constructed binary indicators of occurrence and binary indicators of whether the event occurred before or after the threshold age for brain maturity. Approximately 7% of our sample

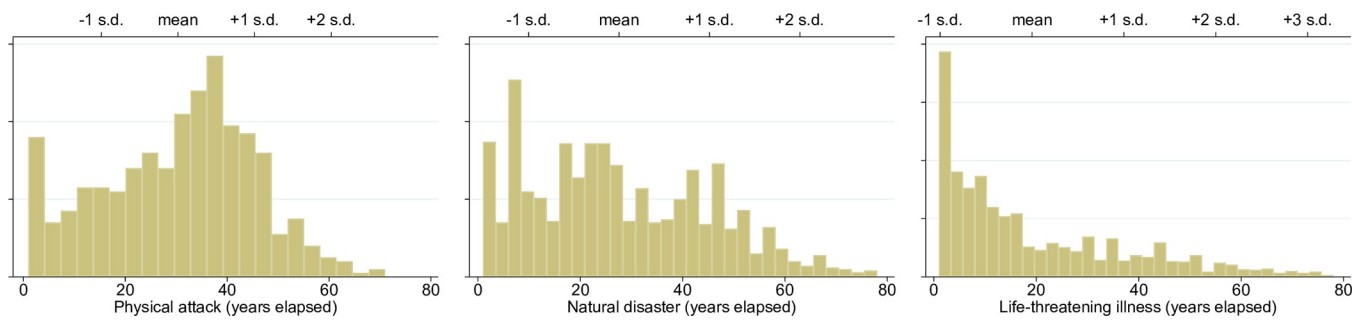

**Fig 1. Frequency histogram of years elapsed after traumatic events.**

reported a physical attack, 17% reported exposure to a natural disaster, and 25% reported suffering from a life-threatening illness. The mean numbers of years elapsed since physical attacks, natural disasters, and life-threatening illnesses were 30.3, 27.6, and 19.2, respectively (see Fig 1). S1 Table presents detailed descriptions of these variables.

**Empirical specification.** We specify the following form of the regression for individual $i$ at time $t \in \{2010, 2012\}$:

$$Y_{i,t} = \alpha_0 + \alpha_1 T_{i,t} + \alpha_2 X_{i,t} + \alpha_3 Z_i + \tau_t + \varepsilon_{i,t}, \tag{1}$$

where $Y_{i,t}$ is a six-point score for self-control; $T_{i,t}$ are variables for the occurrence of traumatic events or the timing of their occurrence; $X_{i,t}$ is a vector of covariates at time $t$; $Z_i$ are indicators of early life conditions; $\tau_t$ are time fixed effects; and $\varepsilon_{i,t}$ is an *i.i.d.* disturbance term. $\tau_t$ denotes dummies for the survey month and the 2012 survey, which represents the average time effects that are common for individuals surveyed at the same time. The regression models were estimated using linear ordinary least squares. To provide an intuitive interpretation of effect sizes, we present the percentage change interpretations using back-of-the-envelope calculations.

$X_{i,t}$ includes age (in years), gender (female or male), race (Black or Hispanic, White, or other races), marital status (married or not in a marital relationship), number of living children, self-rated health (poor or fair, good or better), number of private health insurance plans, out-of-pocket (OOP) medical spending, employment status, household income, total net worth, and census region of residence (Northeast, Midwest, South, West, or other). The OOP medical spending is the amount spent on medical services by the respondent since the last interview. Employment status is coded into three exclusive categories: fully or partially employed, unemployed, and retired or not in the labor force. Household income is the sum of earnings, pensions and annuities, social security benefits, government transfers, capital income, and other income for both spouses. Total net worth is the net value of financial and non-financial household assets, including primary residence, real estate, vehicles, business equity, and retirement accounts. Household income is transformed by the log of household income plus 1. Total net worth is transformed using log-modulus transformation. All dollar figures are in 2012 rates.

The respondents' socioeconomic characteristics around the time they were exposed to specific events are a potentially omitted factor that could confound the estimation of $\alpha_1$. Leaving early life conditions out of the model could lead to a biased estimation of $\alpha_1$ as such conditions may partially determine the well-being outcomes in later life [56]. To address this concern, we expanded our regression with a vector of variables for the respondents' characteristics in childhood and midlife. Specifically, $Z_i$ includes the variables for place of birth (born in the South or other regions), parental educational background (years of education), quality of the

relationship with parents before the age of 18 years, and history of parental substance abuse (S1 Table). It also includes OOP medical spending, employment status, household income, and total net worth obtained from the 1998 survey (HRS, AHEAD, CODA, and WB cohorts) and the 2004 survey (EBB cohort).

## Results

Table 1 presents the average descriptive statistics for the study sample. The sample characteristics are presented for the full sample and for the subsamples divided according to trauma exposure (a sample of 3471 persons who had experienced physical attacks, natural disasters, or life-threatening illnesses and a sample of 5296 persons without such experience). The average respondent was 63.22 years old, had an average of 2.66 living children, owned 0.75 private health insurance plans, and held a net worth of $559,858. Most participants were married (72%) and rated their health condition as good or better (81% good or better, 19% poor or fair). Regarding traumatic life events, 7%, 17%, and 25% of the respondents reported experiencing physical attacks, natural disasters, and life-threatening illnesses, respectively. A comparison of the self-control scores by trauma exposure showed that those who experienced traumatic events had lower self-control than those who did not ($p < 0.05$).

Table 2 presents the main regression results. We begin with the null model, controlling for only trauma indicators, and then gradually expand the model with additional control variables. The coefficient estimates for the covariates are omitted for brevity. Standard errors are clustered at the individual and age levels by using two-way clustering.

**Table 1. Average descriptive statistics ($N = 8767$).**

|  | No trauma (5296 obs.) | Trauma (3471 obs.) | Full sample |
|---|---|---|---|
| Self-control | 4.40 | 4.30[b] | 4.36 |
| Physical attack |  |  | 0.07 |
| Natural disaster |  |  | 0.17 |
| Life-threatening illness |  |  | 0.25 |
| Age | 63.00 | 63.58 | 63.22 |
| Female | 0.57 | 0.51[b] | 0.55 |
| Black or Hispanic | 0.11 | 0.10[b] | 0.10 |
| High school graduate | 0.33 | 0.29[b] | 0.32 |
| Some college | 0.24 | 0.31[b] | 0.27 |
| College and above | 0.34 | 0.34 | 0.34 |
| Married | 0.74 | 0.70[b] | 0.72 |
| No. of living children | 2.68 | 2.62 | 2.66 |
| Poor or fair health | 0.14 | 0.26[b] | 0.19 |
| No. of health insurance | 0.77 | 0.72[b] | 0.75 |
| OOP medical spending[a] | 2735 | 3764[b] | 3137 |
| Employed | 0.45 | 0.37[b] | 0.42 |
| Unemployed | 0.04 | 0.04 | 0.04 |
| Household income[a] | 101,409 | 89,523[b] | 96,760 |
| Total net worth[a] | 585,069 | 520,614[b] | 559,858 |

Health and Retirement Study, 2010–2012. Sample characteristics are weighted by individual- and household-level weights provided in the RAND HRS data.

[a] Dollar figures are adjusted to 2012 dollars using the Consumer Price Index for all urban consumers (CPI-U).

[b] Two-sample $t$-test by traumatic life event rejects the null hypothesis of no difference at the 5% significance level.

**Table 2. Self-control and traumatic life events.**

|  | (1) | (2) | (3) | (4) | (5) | (6) | (7) |
|---|---|---|---|---|---|---|---|
| Physical attack | -0.182*** | -0.162*** | -0.137*** | -0.078*** | -0.058** | -0.066** | -0.103*** |
|  | (0.034) | (0.036) | (0.037) | (0.026) | (0.028) | (0.030) | (0.030) |
| Natural disaster | 0.001 | -0.007 | -0.004 | 0.002 | 0.001 | 0.007 | 0.006 |
|  | (0.025) | (0.026) | (0.026) | (0.020) | (0.019) | (0.020) | (0.020) |
| Life-threatening illness | -0.081*** | -0.041** | -0.037** | -0.017 | -0.037** | -0.003 | -0.019 |
|  | (0.019) | (0.017) | (0.016) | (0.012) | (0.017) | (0.018) | (0.017) |
| Demographic and SES[a] |  | ✓ | ✓ | ✓ | ✓ | ✓ | ✓ |
| Time and region dummies[b] |  | ✓ | ✓ | ✓ | ✓ | ✓ | ✓ |
| Early life controls[c] |  |  | ✓ | ✓ | ✓ | ✓ | ✓ |
| Personality traits[d] |  |  |  | ✓ | ✓ | ✓ | ✓ |
| Matching by attack |  |  |  |  | ✓ |  |  |
| Matching by disaster |  |  |  |  |  | ✓ |  |
| Matching by illness |  |  |  |  |  |  | ✓ |
| Observations | 8767 | 8767 | 8767 | 8741 | 8680 | 8697 | 8684 |

Standard errors in parentheses are clustered at the individual × age levels. Significance levels are indicated by

*, **, and *** for 10, 5, and 1% significance level, respectively.

[a] Demographic and socioeconomic status includes age, gender, race, marital status, number of living children, self-rated health, private health insurance plans, OOP medical spending, employment status, household income, and total net worth.

[b] Time and region dummies include month-of-survey, year-of-survey, and Census region fixed effects.

[c] Early life controls include place of birth and parents' characteristics in earlier waves (education, substance abuse, labor supply, relationship with respondents, OOP medical spending, household income, and total net worth).

[d] Personality traits include conscientiousness, neuroticism, extraversion, openness to experience, and agreeableness.

The null model in column 1 shows a negative correlation between self-control and the experience of a physical attack ($\beta = -0.182$; $p < 0.01$) and a life-threatening illness ($\beta = -0.081$; $p < 0.01$). After adjusting for the baseline covariates and time/region fixed effects (column 2), the coefficient estimates on the two trauma indicators are relatively low but remain negatively significant at the 5% level. A rough comparison with the sample mean of self-control shows that physical attack is associated with a 3.7% decline ($-0.162 / 4.36$) in self-control and that life-threatening illness is associated with a 0.94% reduction ($-0.041 / 4.36$). Further conditioning on early life conditions (column 3) leads to a coefficient estimate of $-0.137$ or a 3.1% decline in self-control in response to attacks ($-0.137 / 4.36$) and an estimate of $-0.037$ or a 0.8% decline in self-control due to illness ($-0.037 / 4.36$). The estimated 3.1% decline is roughly comparable to the effect of being unemployed and is half the effect of having a college degree or being an ethnic minority. The fact that memory from almost 30 years ago explains the influence of education and race bolsters our hypothesis that the influence of traumatic life events is long-lasting.

The regression in column 4 additionally control for measures of the Big Five personality traits (i.e., conscientiousness, neuroticism, extraversion, openness to experience, and agreeableness). This specification allows us to test whether the results from the previous specification are driven by a shift in personality or in self-control skills beyond the respondents' personality. The results show that the estimate on physical attack remains negatively significant ($\beta = -0.078$; $p < 0.01$), whereas the estimate on life-threatening illnesses is no longer different from zero at the 5% level. This result suggests that the influence of physical attacks is not limited to personality shifts but leads to a meaningful reduction in one's ability to regulate impulses.

The subsequent regressions are weighted by the propensity score weight from matching the sample by exposure to a traumatic event. Matching on "observables" has the advantage of balancing out differences in the observed characteristics between the treated (with the traumatic event) and untreated (without traumatic event) groups, thereby reducing bias resulting from nonrandom exposure to a traumatic event. We estimate a logistic regression for exposure to physical attacks (column 5), natural disasters (column 6), or life-threatening illnesses (column 7) and use the kernel matching method with a bandwidth of 0.02. The regressions are adjusted for demographic and socioeconomic characteristics, region/time fixed effects, early life conditions, and personality factors, including other traumatic event indicators not used for matching. Reweighting the regressions with propensity score weights leads to a negative association between physical attacks and self-control at the 5% level (columns 5–7).

Next, we examine whether the association between traumatic events and self-control differs according to exposure timing (Table 3). For each traumatic event, we create a dummy variable for an event that occurred at age $x$ or less (early life exposure) and another for an event that occurred after age $x$ (late-life exposure), where $x$ is the critical age of brain maturity. The age at which the brain stops developing and reaches its peak is not clear, with recent evidence showing that the human brain continues to mature well into the 20s or 30s [57]. The functional magnetic resonance imaging study by Dosenbach et al. [58] revealed that the human brain reaches its adult volume by age 22 and that some regions, such as the frontal lobe, continue to grow past this age until the age of 24 or 32. Currently, no consensus has been reached about the threshold for full brain maturity. Therefore, we use the three age groups suggested by Dosenbach et al. [58] to construct indicators of early- and late-life trauma exposure.

The regressions in Table 3 are conditioned on the baseline covariates and fixed effects, as specified in Eq (1). Across all columns, exposure to a traumatic event is negatively related to self-control regardless of when the event occurred ($p < 0.05$ for all estimates of $\alpha$). The F-tests for the pair of estimates concerning experience at age $\leq x$ vs. experience at age $> x$ fail to reject

**Table 3. Self-control and traumatic life events, by timing of events.**

|  | (1) | (2) | (3) |
|---|---|---|---|
| **Cutoff age:** | **22** | **24** | **32** |
| $\alpha_{1,1}$: Physical attack $\leq$ cutoff age | -0.130** | -0.118** | -0.165*** |
|  | (0.058) | (0.059) | (0.034) |
| $\alpha_{1,2}$: Physical attack $>$ cutoff age | -0.141*** | -0.147*** | -0.108** |
|  | (0.048) | (0.049) | (0.055) |
| Natural disaster $\leq$ cutoff age | -0.003 | -0.022 | -0.029 |
|  | (0.057) | (0.055) | (0.045) |
| Natural disaster $>$ cutoff age | -0.004 | 0.001 | 0.010 |
|  | (0.025) | (0.026) | (0.023) |
| Life-threatening illness $\leq$ cutoff age | -0.030 | -0.019 | -0.017 |
|  | (0.043) | (0.037) | (0.031) |
| Life-threatening illness $>$ cutoff age | -0.038** | -0.040** | -0.041** |
|  | (0.019) | (0.018) | (0.017) |
| Observations | 8767 | 8767 | 8767 |
| Linear restriction (p-value): |  |  |  |
| $H_0$: $\alpha_{1,1}-\alpha_{1,2} = 0$ | 0.89 | 0.71 | 0.28 |

Regressions control for demographic and socioeconomic characteristics, time and region dummies, and early life controls. Standard errors in parentheses are clustered at the individual × age levels. Significance levels are indicated by

*, **, and *** for 10, 5, and 1% significance level, respectively.

Table 4. Health, well-being, and life satisfaction.

| Outcome: | (1) Log(BMI) | (2) Obese | (3) No. of health conditions | (4) CES-D | (5) Exercise | (6) Log (total NW) | (7) life satisfaction |
|---|---|---|---|---|---|---|---|
| Self-control | -0.015*** | -0.035*** | -0.109*** | -0.221*** | 0.010** | 0.200*** | 0.186*** |
| | (0.002) | (0.006) | (0.014) | (0.027) | (0.004) | (0.068) | (0.016) |
| Observations | 8660 | 8767 | 8766 | 8612 | 8767 | 8767 | 8559 |

Regressions control for demographic and socioeconomic characteristics and time and region dummies. Regressions in columns (7) do not include total net worth.
Standard errors in parentheses are clustered at the individual × age levels. Significance levels are indicated by
*, **, and *** for 10, 5, and 1% significance level, respectively.

the null hypothesis at the 5% level (linear restrictions at the bottom of Table 3), indicating that the correlation between physical attacks and self-control does not differ according to exposure timing. Overall, there is no evidence that an attack experienced at a critical age for brain development is significantly related to self-control.

Finally, we evaluate the predictive validity of our self-control measure by examining its relationship with important life outcomes known to be correlated with self-control (Table 4). Following the evidence that self-control predicts health, health behavior, and financial well-being, we estimate the association between self-control and body mass index (BMI), obesity (BMI > 30), number of chronic health conditions, depression (Center for Epidemiologic Studies Depression scale), exercise (one for moderate physical activity and zero for no physical activity), log of total net worth, and life satisfaction. Similar to previous regressions, these models include demographic and socioeconomic covariates and time/region fixed effects. Overall, the results confirm the prior finding that self-control is predictive of health outcomes (lower BMI, less obesity, less chronic disease, and fewer depressive symptoms), health behavior (increased participation in physical activity), wealth accumulation (total net worth), and life satisfaction. All coefficient estimates on self-control are statistically significant at the 5% level and have the expected sign. Although suggestive, these results show that our self-control measure is a valid indicator of self-control capacity over many domains of life outcomes.

## Discussion

This study examined the association between traumatic life events and self-control in old age. Our main results revealed an average 3.1% reduction in self-control in response to a serious physical attack or assault. The association between exposure to attack and self-control did not differ according to whether the attack was experienced before or after the age of 24 years. The mean number of years elapsed since the attack or assault was approximately 30, conditional on exposure, meaning that self-control never fully recovered to the pre-exposure level. For other types of traumatic events (natural disasters and life-threatening illnesses), we found no evidence of shifts in self-control in response to exposure. Our inference regarding the effects of physical attacks and assaults was robust to controlling for early life conditions and using propensity score reweighting to balance out underlying heterogeneity by trauma exposure.

The weak correlations of self-control with natural disasters and life-threatening illnesses might be explained by the strength of the emotional backlash triggered by each event. Shocking and salient events are known to be long-lasting and accessible in memory. Among the three events considered in this study, physical attacks and assaults might have produced the most intense emotions. Though suggestive, our preliminary findings revealed strong associations between physical attacks and assaults with the number of depressive symptoms. This

result raises the possibility that physical attacks and related emotional changes manifest as post-traumatic stress disorder and disrupt the brain's modulatory core.

This study has several limitations. First, the higher mortality in the trauma group raises the concern that our sample may under-represent severely traumatized individuals and over-represent those exposed to relatively low risks. Under the plausible assumption that those excluded from the sample were the most impulsive, addressing survivorship bias would lead to a stronger negative association between trauma exposure and self-control. Second, it remains unclear how traumatic life events produce a lasting influence on self-control. Our findings could be interpreted as the result of mental health problems, such as post-traumatic stress disorder and clinical depression, or financial hardship accompanied by traumatic events. Recent studies have shown that episodes of depressive symptoms are associated with impatience and short-sighted financial planning [59–61]. Future research needs to validate the suggested mechanisms by using alternative data with measurements of possible mediators. Finally, there could be unobserved changes in individual and household characteristics that may have occurred around the trauma exposure. Although childhood covariates and propensity score matching mitigate these effects, we cannot completely rule out the possibility that omitted variables endogenously determine trauma exposure and confound its association with self-control. Future research may consider exploiting exogenous events, such as natural disasters and armed conflicts, to identify changes in self-control attributable to trauma exposure.

Despite these limitations, our findings may serve as a useful guide for future researchers aiming to examine the formation of self-control. Specifically, the demographic, socioeconomic, and trauma indicators used in this study jointly explained less than 40% of an individual's self-control. The limited explanatory power of the estimated models calls for a more rigorous and systematic investigation of how self-control evolves over a lifetime. Behavioral science researchers are encouraged to expand this study by using a multi-item measure of self-control, an exhaustive set of mediating variables, and alternative data that oversample respondents with trauma experience in experimental and observational settings.

As reviewed above, a lack of self-control is related to poor health and economic well-being over the life cycle. Thus, exposure to traumatic life events not only harms mental health of victims, but also incurs significant economic costs. This study suggests that the adverse effects of traumatic life events persist into older age and that the economic cost would be considerable if lifetime consequences were considered. Overall, our findings offer new insight into the potential long-term effects of traumatic life events on self-control. Awareness of these effects may help authorities identify appropriate target cohorts for intervention and formulate policy responses that either directly address self-control problems or improve coping skills. As those at the bottom of the economic scale are more likely to experience traumatic events, the benefits of such interventions would accrue disproportionately to the least advantaged groups.

## Supporting information

**S1 Table. Description of variables.**
(DOCX)

## Author Contributions

**Conceptualization:** Youngjoo Choung.

**Data curation:** Youngjoo Choung, Tae-Young Pak.

**Formal analysis:** Youngjoo Choung, Tae-Young Pak.

**Investigation:** Youngjoo Choung, Tae-Young Pak.

**Methodology:** Youngjoo Choung, Tae-Young Pak.

**Project administration:** Tae-Young Pak.

**Software:** Tae-Young Pak.

**Supervision:** Tae-Young Pak.

**Validation:** Youngjoo Choung, Tae-Young Pak.

**Writing – original draft:** Youngjoo Choung, Tae-Young Pak.

**Writing – review & editing:** Youngjoo Choung, Tae-Young Pak.

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
