## [Decision Letter · Decision Letter 0]

2 Aug 2022

PONE-D-22-08050Traumatic Experience and Self-Control in Old AgePLOS ONE

Dear Dr. Pak,

Thank you for submitting your manuscript to PLOS ONE. After careful consideration, we feel that it has merit but does not fully meet PLOS ONE’s publication criteria as it currently stands. Therefore, we invite you to submit a revised version of the manuscript that addresses the points raised during the review process.

As you can see, Reviewer 2 has several comments on the analysis/empirical strategy/discussion. While I do think they are valid, I also think they can all be solved by an open discussion of these issues and/or some additional regressions being run.I thus strongly encourage you to consider these comments carefully and provide adequate discussions in the paper, for those points that cannot be solved by running additional regressions.

We look forward to receiving your revised manuscript.

Kind regards,

Christiane Schwieren, Dr.

Academic Editor

PLOS ONE

https://journals.plos.org/plosone/s/file?id=ba62/PLOSOne_formatting_sample_title_authors_affiliations.pdf".

2.We note that you have referenced ) Cobb-Clark, D. A., Dahmann, S., Kamhöfer, D., & Schildberg-Hörisch, H. (2019). Self-control: Determinants, life outcomes and intergenerational implication. Unpublished manuscript.) and (Tellegen, A. (1982). Brief manual for the differential personality questionnaire. Unpublished manuscript) which has currently not yet been accepted for publication. Please remove this from your References and amend this to state in the body of your manuscript as detailed online in our guide for authors

http://journals.plos.org/plosone/s/submission-guidelines#loc-reference-style ".

Reviewers' comments:

Reviewer's Responses to Questions

**Comments to the Author**

1. Is the manuscript technically sound, and do the data support the conclusions?

Reviewer #1: Yes

2. Has the statistical analysis been performed appropriately and rigorously? 

Reviewer #1: Yes

3. Have the authors made all data underlying the findings in their manuscript fully available?

Reviewer #1: Yes

4. Is the manuscript presented in an intelligible fashion and written in standard English?

Reviewer #1: Yes

5. Review Comments to the Author

Reviewer #1: It is suggested to include in the title the times of occurrence of the traumatic experiences, early, late or in old age, so that the reader understands a little more what the research is about without having to read the full text, as well as in the introduction , since those are the variables that are being evaluated mainly in the research

It is requested to improve figure 1, it does not have good quality, as well as to identify or rather to differentiate the axes, especially to identify the one that corresponds to age to avoid confusion, especially for new researchers in the area, who wish to continue their research.

6. PLOS authors have the option to publish the peer review history of their article (what does this mean?). If published, this will include your full peer review and any attached files.

Reviewer #1: No

---

## [Author Response · Author response to Decision Letter 0]

19 Aug 2022

The rebuttal letter is attached.

---

## [Decision Letter · Decision Letter 1]

11 Nov 2022

PONE-D-22-08050R1More than just a bad day? Traumatic life events and self-control in old agePLOS ONE

Dear Dr. Pak,

Thank you for submitting your manuscript to PLOS ONE. After careful consideration, we feel that it has merit but does not fully meet PLOS ONE’s publication criteria as it currently stands. Therefore, we invite you to submit a revised version of the manuscript that addresses the points raised during the review process.

 Due to the unavailability of one of the two reviewers of the first round, we had to invite a new reviewer. His comments relate mainly to things that were not clear to him. As he is an experienced researcher in your field, this was an indication for me that some clarification should be done in line with his comments. So I am not asking you to make substantial changes to the paper, but just provide the clarifications that he asks for in his review, to make sure that readers in the field get a very clear understanding of what you are doing and why you are doing this.

We look forward to receiving your revised manuscript.

Kind regards,

Christiane Schwieren, Dr.

Academic Editor

PLOS ONE

Journal Requirements:

Reviewers' comments:

Reviewer's Responses to Questions

**Comments to the Author**

1. If the authors have adequately addressed your comments raised in a previous round of review and you feel that this manuscript is now acceptable for publication, you may indicate that here to bypass the “Comments to the Author” section, enter your conflict of interest statement in the “Confidential to Editor” section, and submit your "Accept" recommendation.

Reviewer #1: All comments have been addressed

Reviewer #2: All comments have been addressed

2. Is the manuscript technically sound, and do the data support the conclusions?

Reviewer #1: Yes

Reviewer #2: Partly

3. Has the statistical analysis been performed appropriately and rigorously? 

Reviewer #1: Yes

Reviewer #2: No

4. Have the authors made all data underlying the findings in their manuscript fully available?

Reviewer #1: Yes

Reviewer #2: Yes

5. Is the manuscript presented in an intelligible fashion and written in standard English?

Reviewer #1: Yes

Reviewer #2: Yes

6. Review Comments to the Author

Reviewer #1: (No Response)

Reviewer #2: This study using HRS data is interesting, because it links the experience of traumatic events rather long time periods ago with current self-control in those 50 years and older. As was found, even traumatic events about 30 years ago seemed to impact of self-control in old age, although effect size seems very limited with an incremental effect of about 3% variance explanation, controlling for relevant confounders. Still, given the long time period, this is kind of a sizable effect, that also might imply applied considerations in terms of prevention and early interventional means. Here are my concerns:

1. At the conceptual level, a range of constructs all assessed in the HRS could be good candidate as dependent variables such as depressed mood, well-being, or attitudes to aging. I found the rationale behind choosing self-control not sufficiently elaborated. This certainly is a key construct in early life or as a ‘life starting’ variable as the Moffitt et al study has convincingly shown. However, I would argue that the construct becomes less important for older adults compared for example with depression. Two request: (1) Please provide a more convincing argument why you chose self-control; (2) what is the difference between self-control and locus of control, which eventually has also been assessed in the HRS?

2. Are you trying to make an argument that early traumatic experiences are also initiate structural or physiological brain impairment that may last for the next decades? If so, please offer a more convincing argument, ideally backed by brain-imaging data. If so, I would expect that cognitive functioning might indeed be a better dependent variable than self-control.

3. I am confused that on p. 3 in the intro part already the findings of the study are described? This is at least totally unusual in my ‘journal world’.

4. Referring to early trauma experiences cannot in my view not be separated from childhood adversities. To my knowledge, linkages among childhood adversities and late-life adverse outcomes have already been researched with the HRS data. Are you indeed heading for the full picture of traumatic experiences, i.e., childhood adversities and what happens later until base line assessment? Or even during the different waves of the HRS running since something like 1993? This needs to crystal-clear in my view.

5. On p. 4 you mention self-control and “associated well-being outcomes”. I agree that this would be important, but which such well-outcomes were included in your study? Does this also mean that you maintain that self-control is a mediator for developmental outcomes? If so, then this needs to be shown empirically.

6. It seems that the full flow of argument is driven by a ‘risk perspective’. However, findings on resilience and adaptational potentials seems to be ignored to a large extent. For example, we know that long-term development comes with habituation and adaptation. Therefore, at least the kind of traumatic events need to be differentiated in those being of very high risk and those of medium or low risk for the rest of lifespan development.

7. There is also talk about traumatic experiences and economic preferences. Is this part of the research aims of the study?

8. Just to make sure: Did you control for self-control level in previous waves or at baseline in your statistical model? Sorry, for having missed this.

9. As a life-span and aging researcher, I see a clear need to differentiate for different life phases in late-life. Controlling for age is not sufficient. For example, there is evidence that early trauma may gain a new momentum in advanced old age, because over vulnerability may impair psychological protection.

7. PLOS authors have the option to publish the peer review history of their article (what does this mean?). If published, this will include your full peer review and any attached files.

Reviewer #1: No

Reviewer #2: No

---

## [Editor Report · Decision Letter 2]

27 Jan 2023

More than just a bad day? Traumatic life events and self-control in old age

PONE-D-22-08050R2

Dear Dr. Pak,

We’re pleased to inform you that your manuscript has been judged scientifically suitable for publication and will be formally accepted for publication once it meets all outstanding technical requirements.

Kind regards,

Christiane Schwieren, Dr.

Academic Editor

PLOS ONE
---

## [Editor Report · Acceptance letter]

31 Jan 2023

PONE-D-22-08050R2 

More than just a bad day? Traumatic life events and self-control in old age 

Dear Dr. Pak:

I'm pleased to inform you that your manuscript has been deemed suitable for publication in PLOS ONE. Congratulations! Your manuscript is now with our production department. 

Kind regards, 

on behalf of

Dr. Christiane Schwieren 

Academic Editor

PLOS ONE